# Programmable Exploration of Synthesizable Chemical Space

## Abstract

The constrained nature of synthesizable chemical space poses a significant challenge for sampling molecules that are both synthetically accessible and possess desired properties. In this work, we present a programmable model for the discovery of molecules within synthesizable space. The model can generate synthesizable molecules subject to complex logical queries of molecular properties. By leveraging this querying capability, molecular optimization with respect to black-box oracle functions can be performed through iterative refinement of the queries, which achieves high optimization efficiency while preserving synthesizability. We demonstrate the model's high coverage of the synthesizable chemical space, achieving a record-high 92% reconstruction rate on a chemical space projection test set curated from the Enamine REAL database. We then evaluate its capability for complex query-based molecular design through a series of multi-objective molecular discovery benchmarks. Finally, we show that the query-based molecular optimization technique achieves substantially higher sampling efficiency than both synthesis-based and synthesis-agnostic methods.

## 1 Introduction

Advances in generative models have led to the proliferation of new methods for molecular design, offering higher sampling efficiency than enumerative virtual screening (Gómez-Bombarelli et al., 2018; Gao et al., 2022). Predominantly, generative models represent molecules in the form of strings, graphs, or 3D coordinates, which are agnostic of synthesizability. As a result, these models tend to propose molecules that are difficult or even impossible to synthesize in practice (Gao & Coley, 2020). The lack of synthesizability guarantee has been a major bottleneck hindering experimental validation and translation to biomedical applications.

Recently, methods that focus on generating synthetic pathways rather than unconstrained molecular graphs have been developed to address the synthesizability issue. Among these, one notable category is the "chemical space projection" approach which aims to find structurally similar molecules, or analogs, within the synthesizable space for any molecular graph (Luo et al., 2024). In this paradigm, an external generative model is first used to draft molecular graphs that satisfy specific properties but are not necessarily synthesizable. Then, these molecular graphs are "projected" to synthesizable analogs in form of *postfix notation of synthesis*, a linear and synthetic pathway-based molecular representation. This two-stage process inevitably creates structural and functional inconsistencies between the molecular graphs and the synthesizable analogs, resulting in compromised properties. Therefore, it is crucial to develop models that operate directly within the synthesizable chemical space to unify property and synthesizability.

Directly designing molecules in the synthesizable chemical space has been explored in recent works (Gottipati et al., 2020; Swanson et al., 2024; Cretu et al., 2024), yet their performance remains limited due to several unaddressed challenges. First, the action space of building block selection is extremely large, the size of which often reaches hundreds of thousands or millions (Enamine, 2025). Existing methods generate synthetic pathways by explicitly selecting building blocks from such large libraries, which is difficult to sufficiently explore chemical space during both training and sampling. Second, the search space of synthetic pathways is sparse as not all combinations of building blocks and reactions lead to valid syntheses. Lastly, the representation of synthetic pathways is not informed of the structural features of the resulting product molecules. Models that

learn distributions over synthetic pathways are typically only aware of the combinatorial rules of reactions and building blocks. As a result, the structural features of the products, which are crucial determinants of molecular properties, are not captured, leading to a gap between pathway generation and property optimization. These issues limit the sampling efficiency of existing methods, making them even less practical for real-world applications that require expensive property evaluations or involve multiple property constraints.

Given the above challenges, we ask the following question: can we develop a generative model that (1) guarantees synthesizability, (2) generates molecules directly within the synthesizable chemical space to preserve molecular properties, (3) generates molecules subject to multiple property constraints, and (4) has high sampling efficiency for black-box oracle functions?

As a solution, we introduce **PrexSyn**[1], a generative framework for synthesizable molecular design. To ensure synthesizability, we use the postfix notation of synthesis (Luo et al., 2024). Unlike prior works that generate synthetic pathways according to molecular graphs, we design a transformer language model that directly generates postfix notations of synthesis conditioned on a variety of property prompts. To enable logical composition of properties (Du et al., 2020), *i.e.* multiple properties connected by logical operators (AND, NOT, OR), we develop a sampling algorithm that compiles logical queries into arithmetic combinations of probability distributions conditioned on each individual property prompt, allowing users to "program" generation objectives. Further, this querying capability also serves as an interface for optimizing molecules with respect to black-box oracle functions *in the query space* via iterative refinement of property queries based on oracle feedback. This new optimization paradigm avoids direct selection from the vast building block action space and bridges the gap between synthesis and property representations, enabling efficient exploration of the synthesizable chemical space.

Our model is trained on pairs of synthetic pathways and their corresponding molecular properties. However, the term "molecular property" encompasses a wide range of molecular characteristics, including structure, protein binding, biological activity, and so on — it is impossible to enumerate all such properties for model training. Therefore, in our work, only a set of efficiently-computable *structural properties* are used during training, including molecular fingerprints, scaffold structures, physicochemical descriptors, pharmacophoric features, and so on. This design choice is motivated by the structural-functional principle, which states that the structure of a molecule determines its function. Consequently, *functional properties* can be characterized through structural properties. By focusing on an essential set of structural properties, the model is adaptable to functional properties during inference.

We demonstrate the effectiveness of our approach through extensive experiments. First, our model achieves high coverage of the synthesizable chemical space, achieving a record-high 92% reconstruction rate on a chemical space projection test set curated from the Enamine REAL database. Second, it can generate synthesizable molecules that satisfy complex logical queries involving multiple properties. Third, it enables efficient molecular optimization with respect to black-box oracle functions, requiring fewer oracle calls to achieve comparable or even better performance than baseline methods.

## 2 METHOD

In this section, we first describe the model architecture of our method in Section 2.1. Then, we present the sampling algorithms for generating molecules satisfying complex property queries in Section 2.2 and for query-based molecular optimization with respect to black-box oracle functions in Section 2.3.

### 2.1 MODEL ARCHITECTURE

The model is a decoder-only transformer (Vaswani et al., 2017) which takes as input a prompt sequence of molecular properties and then autoregressively generates postfix notations of synthesis (Figure 1a). Formally, let $C = [c_1, \ldots, c_M]$ be the embedding vectors of the property prompt, $s = [s_1, \ldots, s_N]$ be the tokenized postfix notation sequence. The model learns the conditional

---

[1]The name is derived from the title ***Pr**ogrammable **Ex**ploration of **Syn**thesizable Chemical Space*.

distribution of $s$ given the property prompt as:

$$p(\boldsymbol{s}|\boldsymbol{C}) = \prod_{i=1}^{N} p(s_i|s_{<i}, \boldsymbol{C}) \tag{1}$$

The transformer architecture allows for flexible handling of various molecular properties, which can take diverse forms, including categorical values, scalar values, vectors, and sequences. Categorical properties are embedded using standard embedding lookup tables. Scalar and vector properties are transformed into fixed-length embeddings using MLPs. For sequential properties, each element is first embedded as a scalar or categorical property, and the resulting sequence of vectors is then fed into the transformer model.

The postfix notation of synthesis is represented as a sequence of tokens, with each token associated with a learnable embedding vector, following the standard practice in transformer language models. Positional encodings are added to the token embeddings to indicate the order of the tokens.

To predict the next token, the model adopts a two-level approach. First, it predicts the class of the next token (*i.e.*, building block, reaction, [START], or [END]). If the predicted class is a building block or reaction token, a second-level classifier is used to predict the specific token within that class. This two-level classifier is similar to the routing mechanism in mixture-of-expert models (Eigen et al., 2013), as it avoids retrieving building blocks at every step, whose computational cost is not insignificant given the large library size. Formally, the conditional distribution of the next token is given by:

$$p(s_i|s_{<i}, \boldsymbol{C}) = \begin{cases} p\left(c(s_i) = \text{BB} \mid \cdots\right) p\left(s_i \mid c(s_i) = \text{BB}, \cdots\right) & c(s_i) = \text{BB} \\ p\left(c(s_i) = \text{RXN} \mid \cdots\right) p\left(s_i \mid c(s_i) = \text{RXN}, \cdots\right) & c(s_i) = \text{RXN}, \\ p\left(s_i \mid \cdots\right) & \text{otherwise} \end{cases} \tag{2}$$

where $c(s_i)$ denotes the class of the token $s_i$. Note that we use a classifier to select building blocks from the library, with each class corresponding to a specific building block. This differs from previous models, which rely on molecular fingerprints to retrieve building blocks via deterministic nearest-neighbor search. Fingerprint-based selection is limited to structural similarity and is not adaptable to varying contexts. In contrast, the classifier-based approach allows dynamic selection of building blocks based on the context provided by property prompts through the learnable unembedding matrix. In addition, it naturally allows probabilistic sampling from the building block library, which is crucial for generating diverse synthetic pathways. While this leads to the introduction of a large number of distinct tokens (*i.e.*, hundreds of thousands), the scalability of the architecture allows each building block to be observed sufficiently during training.

To train the model, we randomly construct synthetic pathways by iteratively selecting and combining building blocks and reaction templates from the library. These pathways are then paired with the properties of their product molecules to form training data. The model is trained using standard cross-entropy loss to maximize the likelihood of the postfix notation sequence conditioned on their property prompts.

At inference time, to generate molecules conditioned on a single property prompt, we first embed the prompt and prepend the embedding vectors to the [START] token. Then, we autoregressively sample tokens until the [END] token is emitted. Finally, we use the synthesis stack simulator (Luo et al., 2024) to virtually execute the postfix notation of synthesis with RDKit (RDKit, 2010) and obtain the product molecular graph.

## 2.2 SAMPLING MOLECULES SUBJECT TO COMPLEX PROPERTY QUERIES

Multiple property conditions are often required in practical scenarios. For instance, one may aim to design molecules that both bind to a specific target and exhibit sufficient solubility, or to generate molecules that share similar pharmacophores to a known compound while having different scaffolds.

In this section, we formulate the composition rule of multiple property prompts (Figure 1b). We assume that the property prompts are mutually conditionally independent given the molecule, and that the underlying prior distribution over molecules is uniform. Under these assumptions, the joint conditional distribution can be expressed in a product-of-experts form, which provides the basis of our formulation (Hinton, 1999). While some structural properties are considered independent,

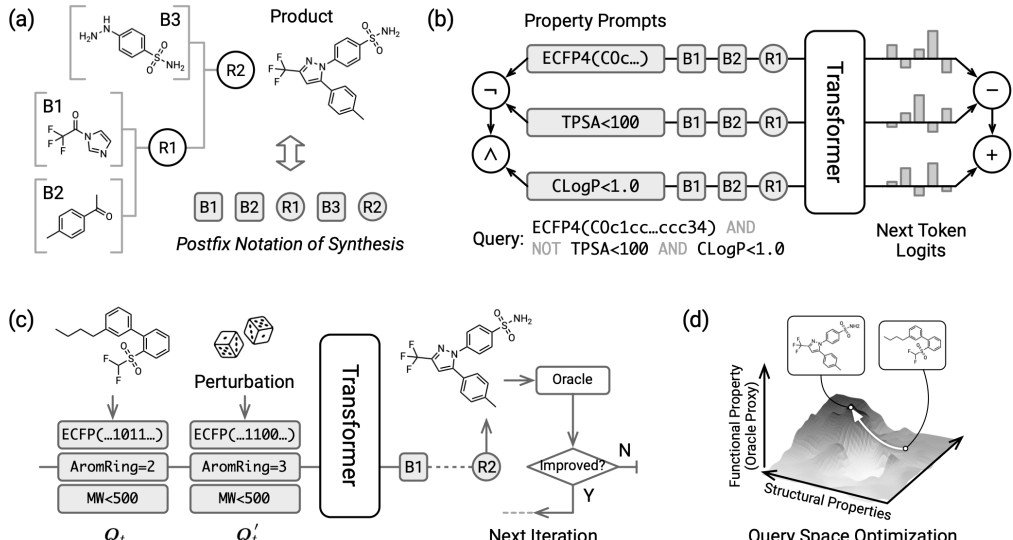

Figure 1: **(a)** Molecules are represented using postfix notation of synthesis. **(b)** The transformer predicts the next token conditioned on property prompts. For composite conditions, token probabilities from separate prompts are combined. **(c)** Query space optimization. At each step, molecular properties are computed and perturbed to recondition the model. Newly generated molecules are then evaluated by oracle functions, and those with improved properties replace previous ones. **(d)** The principle that functional properties can be characterized through structural properties underlies the model's ability to generalize to functional properties at inference time.

such as scaffold and decoration, this assumption is not strictly true, as many molecular properties are correlated, particularly those related to size such as molecular weight and number of rotatable bonds. Nevertheless, this assumption, as a simplification, allows us to derive a practical algorithm that we empirically demonstrate to be highly effective in later sections.

First, consider two different property prompts: $C_1$ and $C_2$; $p(s|C_1)$ and $p(s|C_2)$ are the distributions of postfix notations conditioned on each prompt respectively.

**Conjunction (AND)** The conditional distributions for molecules satisfying both conditions $C_1$ and $C_2$ is the product of the two distributions:

$$p(s|C_1 \wedge C_2) \propto p(s|C_1)^\alpha p(s|C_2)^\beta = \prod_{i=1}^N p(s_i|s_{<i}, C_1)^\alpha p(s_i|s_{<i}, C_2)^\beta \quad (\alpha, \beta > 0), \quad (3)$$

where $\alpha$ and $\beta$ are hyperparameters that control the relative importance of each condition. At each sampling step, we first compute the conditional distributions of the next token given each property prompt, and then combine them using the above equation to get the final distribution for sampling. Specifically, the combined distribution is given by:

$$p(s_i|s_{<i}, C_1)^\alpha p(s_i|s_{<i}, C_2)^\beta = \text{softmax}\left(\alpha z_1 + \beta z_2\right), \quad (4)$$

where $z_1$ and $z_2$ are the logits of the next token predicted by the model given property prompts $C_1$ and $C_2$ respectively.

**Negation (NOT)** Similarly, the conditional distribution for molecules satisfying $C_1$ but not $C_2$ is given by:

$$p(s|C_1 \neg C_2) \propto \frac{p(s|C_1)^\alpha}{p(s|C_2)^\beta} = \prod_{i=1}^N \frac{p(s_i|s_{<i}, C_1)^\alpha}{p(s_i|s_{<i}, C_2)^\beta} \quad (\alpha, \beta > 0). \quad (5)$$

The combined distribution for each sampling step is given by:

$$p(s_i|s_{<i}, C_1 \neg C_2) \propto \frac{p(s_i|s_{<i}, C_1)^\alpha}{p(s_i|s_{<i}, C_2)^\beta} = \text{softmax}\left(\alpha z_1 - \beta z_2\right). \quad (6)$$

Note that the negation operation can be unified with the conjunction operation by allowing $\beta$ in Equation 3 to take negative values, which provides a convenient formulation for implementation.

**Disjunction (OR)** Under the disjunction of two conditions $C_1$ and $C_2$, the distribution of molecules satisfying either condition is given by:

$$p(s|C_1 \vee C_2) \propto \alpha p(s|C_1) + \beta p(s|C_2) = \alpha \prod_{i=1}^{N} p(s_i|s_{<i}, C_1) + \beta \prod_{i=1}^{N} p(s_i|s_{<i}, C_2). \quad (7)$$

Unlike the previous two cases, this distribution cannot be factorized autoregressively. Therefore, we sample full sequences from each conditional distribution separately and then merge the samples to obtain the final set of molecules.

**Complex Logical Queries** We define *query* ($Q$) as a logical expression over molecular properties composed using the logical operators AND, NOT, and OR. To enable sampling from complex logical queries involving multiple properties connected by logical operators, we first convert the query into disjunctive normal form (DNF) (Rosen, 2019), *i.e.* a series of ORs where each term only contains ANDs and NOTs. For each conjunctive term, we apply Equation 4 and 6 to get its composed distribution, from which samples are drawn. Finally, we merge the samples from all conjunctive terms to obtain the final set of molecules satisfying the complex logical query.

In our implementation, supported properties include: (1) topological fingerprint ECFP4 (Rogers & Hahn, 2010); (2) pharmacophoric features (Gobbi & Poppinger, 1998); (3) scaffold structures (Bemis & Murcko, 1996); (4) substructures (fragments); (5) physicochemical descriptors including molecular weight, ClogP, TPSA, *etc*. As discussed earlier in Section 1, the choice of structural properties is motivated by the structural-functional principle, which states that a molecule's structure determines its function. Functional properties such as protein binding and biological activity can be characterized through structural properties (Figure 1d); empirically, note that fingerprint-based and descriptor-based representations of structures achieve competitive performance in quantitative structure-property relationship modeling. Therefore, we expect and later show that the model is adaptable to functional properties at inference time. In the next section, we demonstrate how this querying capability can be used to sample molecules with respect to general properties defined through black-box oracles.

### 2.3 MOLECULAR OPTIMIZATION IN QUERY SPACE

The querying capability provides an interface for optimizing molecules with respect to black-box oracle functions through iterative refinement of property queries guided by oracle feedback (Figure 1c).

The optimization process begins with a seed query $Q_0$, which is used to generate an initial set of candidate molecules. At each iteration, each candidate molecule $\mathcal{M}$ is mapped to a property query $Q_t$. In general, the property query takes the form $Q = C_1^{(\text{opt})}(\mathcal{M}) \wedge C_2^{(\text{opt})}(\mathcal{M}) \wedge \cdots C_1^{(\text{cstr})} \wedge \cdots$, where $\{C_i^{(\text{opt})}(\mathcal{M})\}$ denotes optimizable conditions dependent on $\mathcal{M}$ and $\{C_i^{(\text{cstr})}\}$ denotes fixed constraints. Next, noise is added to the optimizable term, resulting in a perturbed query $Q' = C_1'^{(\text{opt})} \wedge \cdots C_1^{(\text{cstr})} \wedge \cdots$. The perturbed query is then used to generate new candidate molecules, which are evaluated by the oracle. New candidates that achieve a better oracle score will replace the old candidates, leading to an improved set of molecules.

### 2.4 EXPERIMENT SETUP

We used the reaction template set curated by Gao et al. (2024), which contains 115 reaction templates. Following previous works (Luo et al., 2024; Gao et al., 2024; Lee et al., 2025), we used Enamine US in-stock building block set retrieved on October 1, 2023, which contains 223,244 building blocks after RDKit preprocessing. More implementation details are provided in Appendix B. The code and data of this project will be available upon publication.

## 3 RESULTS

### 3.1 CHEMICAL SPACE PROJECTION

We first evaluate our model on the chemical space projection task. This task involves finding synthesizable analogs for given molecular graphs. Two benchmark datasets with different emphases (Luo et al., 2024; Gao et al., 2024) are used for evaluation: (1) **Enamine testset**: 1,000 molecules

Table 1: Chemical space projection results. PrexSyn achieves the highest accuracy and the highest efficiency.

| Method | Enamine REAL | | ChEMBL | | |
| | Recons.% | Similarity | Recons.% | Similarity | Time/Target |
|---|---|---|---|---|---|
| SynNet (Gao et al., 2021) | 11.0% | 0.57 | 5.4% | 0.43 | - |
| SynthesisNet (Sun et al., 2024) | - | - | 9.2% | 0.53 | - |
| ChemProjector (Luo et al., 2024) | 46.0% | 0.81 | 13.0% | 0.60 | 5.15s±4.58s |
| SynFormer (Gao et al., 2024) | 66.0% | 0.91 | 20.0% | 0.67 | 3.45s±3.60s |
| SynLlama (Sun et al., 2025) | 69.1% | 0.92 | 19.7% | 0.68 | 16.21s±9.43s |
| ReaSyn (Lee et al., 2025) | 76.8% | 0.95 | 22.9% | 0.69 | 19.71s±6.68s |
| PrexSyn (beam size=8) | 89.8% | 0.97 | 22.5% | 0.71 | **0.18s**±0.02s |
| PrexSyn (beam size=16) | 92.0% | 0.98 | 24.8% | 0.73 | 0.35s±0.03s |
| PrexSyn (beam size=32) | **92.9%** | **0.98** | **27.3%** | **0.74** | 1.00s±0.06s |

curated from the Enamine REAL database, used to assess how well the model covers the synthesizable chemical space. (2) **ChEMBL testset**: 1,000 molecules from ChEMBL (Gaulton et al., 2012), which are not necessarily synthesizable with the Enamine building blocks and reactions, used to test the model's ability to find synthesizable analogs for arbitrary molecular graphs.

To run the projection task, we first compute the ECFP4 fingerprint of each molecule in the testset and embed it into prompt vectors, which condition the model to generate postfix notations of synthesis. Beam search with beam sizes of 8, 16 and 32 is used for generation. The molecule with the highest fingerprint similarity to the input molecule is selected as the projection result in accordance with the evaluation procedure of previous studies. All inference is conducted on a single NVIDIA 4090 GPU.

As shown in Table 1, our model significantly surpasses previous methods on both datasets in terms of quality and efficiency. In particular, our model achieves a recording-breaking reconstruction rate of 92.9% on the Enamine testset, along with a Tanimoto similarity score over Morgan fingerprints of 0.98, whereas the previous best model only achieves a reconstruction rate of 69.1% and a similarity score of 0.92. This result shows that our model nearly perfectly covers the synthesizable chemical space defined by the Enamine building blocks and reactions, thereby establishing a strong foundation for more complex chemical space exploration tasks. On the ChEMBL testset, our model achieves a reconstruction rate of 27.3% and a similarity score of 0.74, setting a new state-of-the-art performance, which demonstrates its strong capability to find similar analogs for molecules beyond the defined chemical space. In addition to the improved performance, our model is substantially more efficient than previous methods, taking an average of only 0.18 seconds per target with a beam size of 8 and 1.00 seconds with a beam size of 32.

## 3.2 COMPLEX PROPERTY QUERYING

**Overview** We design six query tasks that reflect real-world drug discovery scenarios where complex property constraints are involved, as summarized in Table 2. For each task, we generate 1,000 postfix notations and score the product molecules according to how well they satisfy the specified query. We report the mean and standard deviation of both the average score and the diversity of the top 5% and top 10% highest-scoring molecules across 5 independent runs. Diversity is quantified as 1 minus the average pairwise Tanimoto similarity between Morgan fingerprints of the generated molecules (Jin et al., 2020).

**Task 1** evaluates the generation of drug-like molecules that satisfy Lipinski's Rule of Five (Lipinski, 2004; Chagas et al., 2018), expressed as a conjunction of multiple property constraints. Molecules are scored between 0 and 1 according to the fraction of conditions satisfied. The generated set achieves an average score of 0.9549, with the top 5% and 10% of molecules achieving perfect scores of 1.0000. These top samples also maintain high diversity (above 0.89), indicating that the model produces a broad and varied set of Lipinski-compliant molecules.

**Tasks 2 and 3** are inspired by the GuacaMol benchmarks (Brown et al., 2019), which involve finding analogs of existing drugs with modified physicochemical properties. Generated molecules are evaluated using the corresponding GuacaMol scoring functions, ranging from 0 to 1, with higher scores indicating better satisfaction of the desired properties. On the Cobimetinib optimization benchmark (Task 2), our best molecule achieves a score of 0.9326, with the top 5% and 10% av-

Table 2: Composite property querying results.

| # | Task Description | Query | Best Score | Average Score (↑) | | | Diversity (↑) | |
|---|---|---|---|---|---|---|---|---|
| | | | | T5% | T10% | All | T5% | T10% |
| 1 | Generate molecules that satisfy Lipinski's Rule of 5. | MW<500 AND Donors<5 AND Acceptors<10 AND RotatableBonds<10 AND TPSA<140 AND CLogP<5.0 | 1.0000 ±.0000 | 1.0000 ±.0000 | 1.0000 ±.0000 | 0.9549 ±.0036 | 0.8902 ±.0011 | 0.8902 ±.0011 |
| 2 | Find analogs of Cobimetinib that have 3 rotatable bonds and 3 aromatic rings. Crippen logP should not exceed 5.0 | ECFP4("OC1(CN...") AND RotatableBonds=3 AND AromaticRings=3 AND CLogP<5.0 | 0.9326 ±.0040 | 0.8975 ±.0013 | 0.8848 ±.0017 | 0.7108 ±.0060 | 0.6017 ±.0113 | 0.6770 ±.0176 |
| 3 | Reduce the lipophilicity of Osimertinib by increasing TPSA to above 100 and reducing logP to below 1.0 | ECFP4("COc1cc...") AND NOT TPSA<100 AND CLogP<1.0 | 0.9164 ±.0217 | 0.8314 ±.0081 | 0.8068 ±.0047 | 0.4971 ±.0085 | 0.7499 ±.0234 | 0.8024 ±.0061 |
| 4 | Perform scaffold hopping for a CDK6 inhibitor. (PDB:2EUF, CCD:LQQ) | FeatureSim("LQQ") AND NOT ScaffoldSim("LQQ") | 0.6597 ±.0213 | 0.5977 ±.0077 | 0.5843 ±.0059 | 0.5038 ±.0029 | 0.6658 ±.0548 | 0.6976 ±.0379 |
| 5 | Perform scaffold hopping for a TGFβ1 inhibitor. (PDB:6B8Y, CCD:D0A) | FeatureSim("D0A") AND NOT ScaffoldSim("D0A") | 0.6610 ±.0106 | 0.6164 ±.0018 | 0.6038 ±.0018 | 0.5149 ±.0025 | 0.8075 ±.0044 | 0.8121 ±.0069 |

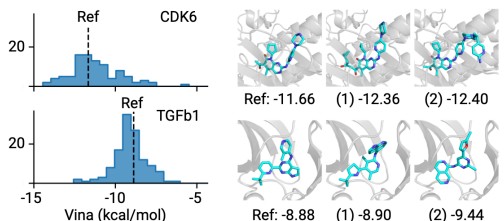

Figure 2: Scaffold hopping query results.

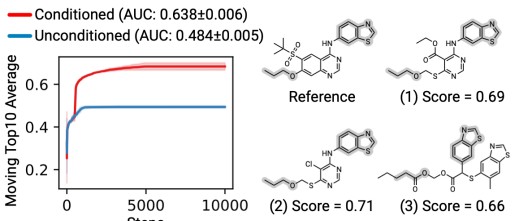

Figure 3: Scaffold hopping optimization results.

eraging 0.8975 and 0.8848, respectively. For comparison, a graph-based approach that explicitly optimizes the four conditions, rather than treating the scoring function as a black box, reported a top score of 0.93 (Verhellen, 2022). On the Osimertinib optimization benchmark (Task 3), our best molecule achieves a score of 0.9164, with the top 5% and 10% averaging 0.8314 and 0.8068, respectively. While the strongest baseline reported by Brown et al. (2019) achieved a slightly higher top score of 0.95, their method directly modifies molecular graphs without guaranteeing synthesizability. Note that this evaluation treats GuacaMol objectives as whitebox desiderata rather than blackbox oracles. Visualizations of the top molecules are provided in Table A1.

**Tasks 4 and 5** focus on identifying molecules that preserve key pharmacophore features of a known drug while adopting a different scaffold, a widely used strategy in drug discovery known as scaffold hopping (Böhm et al., 2004; Schneider et al., 2006). Molecules are scored according to $\frac{1}{2}(\text{sim}_{\text{Gobbi}} + 1 - \text{sim}_{\text{Scaffold}})$, where $\text{sim}_{\text{Gobbi}}$ measures pharmacophore feature similarity using the Gobbi fingerprint (Gobbi & Poppinger, 1998), and $\text{sim}_{\text{Scaffold}}$ quantifies scaffold similarity based on the Murcko scaffold (Bemis & Murcko, 1996). On the CDK6 inhibitor scaffold hopping task (Task 4), the best molecule achieves a score of 0.6597, while the top 5% and 10% of molecules average 0.5977 and 0.5843, respectively. On the TGFβ1 inhibitor scaffold hopping task (Task 5), the best molecule achieves a score of 0.6610, with the top 5% and 10% averaging 0.6164 and 0.6038, respectively.

### 3.3 OPTIMIZATION IN QUERY SPACE

**Composite Query-Based Optimization** We design a scaffold hopping task to demonstrate the optimization capability with respect to composite queries. Given a reference molecule, the goal of this task is to generate molecules that (1) have the same key substructures, (2) have a different scaffold, and (3) share similar pharmacophore features. These goals are wrapped into a scoring function consisting of three terms as the objective.

At each optimization iteration, the optimizable term $C^{(\text{opt})}$ of the query is defined as the scaffold fingerprint and the constraint term is to require the two key substructures, i.e. $C^{(\text{cstr})} = \text{Substruct}(D_1) \wedge \text{Substruct}(D_2)$. The initialization query is Lipinski's Rule of 5 to seed an initial set of random drug-like molecules. We also run a baseline that directly optimizes the fin-

Table 3: GuacaMol benchmark results measured by AUC-Top10 (Gao et al., 2022). PrexSyn achieves the highest sampling efficiency on 6 out of 8 targets while maintaining synthesizability.

| Method | Syn. | Amlo. | Fexo. | Osim. | Peri. | Rano. | Sita. | Zale. | Cele. |
|---|---|---|---|---|---|---|---|---|---|
| REINVENT (Olivecrona et al., 2017) | ✗ | 0.635 | 0.784 | 0.837 | 0.537 | 0.760 | 0.021 | 0.358 | 0.713 |
| GraphGA (Jensen, 2019) | ✗ | 0.651 | 0.785 | 0.829 | 0.533 | 0.745 | 0.524 | 0.458 | 0.682 |
| MolGA (Tripp & Hernández-Lobato, 2023) | ✗ | 0.688 | 0.825 | 0.844 | 0.547 | 0.804 | **0.582** | 0.519 | 0.567 |
| DoG-Gen (Bradshaw et al., 2020) | ✓ | 0.537 | 0.697 | 0.776 | 0.475 | 0.712 | 0.048 | 0.123 | 0.466 |
| SynNet (Gao et al., 2021) | ✓ | 0.567 | 0.764 | 0.797 | 0.559 | 0.743 | 0.026 | 0.341 | 0.443 |
| SyntheMol (Swanson et al., 2024) | ✓ | 0.004 | 0.703 | 0.823 | 0.013 | 0.767 | 0.000 | 0.000 | 0.527 |
| SynthesisNet (Sun et al., 2024) | ✓ | 0.608 | 0.791 | 0.810 | 0.524 | 0.741 | 0.313 | **0.528** | 0.582 |
| SynFormer (Gao et al., 2024) | ✓ | 0.696 | 0.786 | 0.816 | 0.530 | 0.751 | 0.338 | 0.478 | 0.559 |
| ReaSyn (Lee et al., 2025) | ✓ | 0.678 | 0.788 | 0.820 | 0.560 | 0.742 | 0.342 | 0.492 | 0.754 |
| PrexSyn | ✓ | **0.781** ±.023 | **0.837** ±.013 | **0.855** ±.007 | **0.714** ±.010 | **0.807** ±.009 | 0.471 ±.030 | 0.504 ±.018 | **0.801** ±.005 |

gerprint of the full molecule without the decoration conditions. As shown in Figure 3, the composite query-based optimization (red curve) achieves higher efficiency than the condition-free baseline (blue curve). This result demonstrates the composite query's ability to narrow down the search space in scenarios where partial information is available, leading to more efficient optimization. Illustrations of two different optimization landscapes are presented in Figure A1.

**GuacaMol Benchmark**  To quantify the general sampling efficiency of our model, we conduct evaluation on seven multiproperty objectives and one rediscovery task from the GuacaMol benchmark suite (Brown et al., 2019). In this setting, these scoring functions are treated as black boxes, which means no information about their internal form is visible to the model — only the final scores are provided. We set the initialization query of the $Q_0$ to Lipinski's Rule of 5 to generate drug-like molecules as starting candidates. The optimizable term $C^{(opt)}$ is defined as the ECFP4 fingerprint and no constraint term is applied, $C^{(cstr)} = \varnothing$. Genetic algorithm is used to optimize $C^{(opt)}$.

We compare our method with both synthesis-agnostic and synthesis-based baselines. All methods are allowed a budget of 10,000 oracle calls, and AUC-Top10 scores are reported following previous studies (Gao et al., 2022). As shown in Table 3, our method achieves the highest average score across 6 out of 8 tasks. Notably, on Amlodipine MPO and Perindopril MPO, our method significantly improves the best scores from 0.696 to 0.781 and from 0.559 to 0.714 respectively.

### 3.4 Applications in Docking-Based Molecule Design

**sEH**  We evaluate our model on the task of generating ligands for soluble epoxide hydrolase (sEH). The oracle function is defined as the negative docking score predicted by a proxy models trained on molecules docked with AutoDock Vina against the sEH protein structure (Cretu et al., 2024; Bengio et al., 2021). Following Cretu et al. (2024), the predicted scores are normalized by a factor of 1/8.

The setting of the baseline method SynFlowNet (Cretu et al., 2024) differs slightly from ours. SynFlowNet is trained to learn a distribution of molecules with high binding affinity, whereas we focus on directly optimizing binding affinity starting from random molecules. Once trained, SynFlowNet can generate molecules in a single forward pass but requires extensive training data and oracle calls (5000 steps, batch size 64, totaling ∼300k samples). In contrast, our method requires multiple optimization iterations but does not rely on any training samples. While the two settings are not directly comparable, we can still view our method as a sampler with warm-up steps and compare the quality of generated molecules under a stricter oracle budget. Specifically, we allow our model 10,000 oracle calls, about 30 times fewer than SynFlowNet, and evaluate performance using either the final or the top 1,000 generated molecules.

According to Table 4, our method achieves a mean sEH score of 1.01 when selecting the top 1,000 molecules, significantly outperforming SynFlowNet's best-reported score of 0.94. Note that since the score is defined as the negative binding energy divided by 8, values above 1.0 are possible. In addition, our generated molecules achieve better drug-likeness, with a QED score of 0.80 compared to SynFlowNet's 0.68, and an improved SA score of 2.23 versus SynFlowNet's 2.67. While we include the SA score (Ertl & Schuffenhauer, 2009) in the comparison for completeness, we note that SA scores are less relevant in the context of synthesizable molecular design, as providing a synthetic pathway composed only of purchasable building blocks and reaction templates, which both methods do, is a much stronger evidence of synthesizability than the heuristic SA score.

| Method | Syn. | sEH(↑) | SA(↓) | QED(↑) |
|---|---|---|---|---|
| FragGFN | ✗ | 0.77±0.01 | 6.28±0.02 | 0.30±0.01 |
| FragGFN(SA) | ✗ | 0.70±0.01 | 5.45±0.05 | 0.29±0.01 |
| SyntheMol | ✓ | 0.64±0.01 | 3.08±0.01 | 0.63±0.01 |
| SynFlowNet | ✓ | 0.92±0.01 | 2.92±0.01 | 0.59±0.02 |
| SynFlowNet(SA) | ✓ | 0.94±0.01 | 2.67±0.03 | 0.68±0.01 |
| SynFlowNet(QED) | ✓ | 0.86±0.03 | 4.02±0.26 | 0.74±0.04 |
| PrexSyn(Last) | ✓ | 0.85±0.01 | 2.28±0.07 | 0.79±0.00 |
| PrexSyn(Top) | ✓ | 1.01±0.00 | 2.23±0.04 | 0.80±0.01 |

Table 4: Comparison on sEH binding, SA score, and QED score. PrexSyn achieves the best overall performance while ensuring synthesizability.

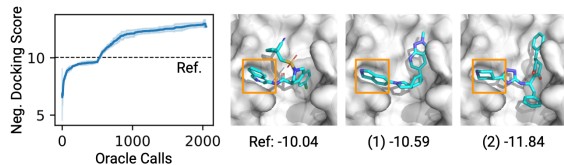

Figure 4: Left: Best-10 average negative docking score vs. oracle calls; dashed line marks the reference inhibitor. Right: Docked poses of the reference inhibitor and two generated molecules with improved scores.

**Mpro2** We further evaluate our model on the task of generating ligand candidates for the SARS-CoV-2 main protease (Mpro2) using AutoDock-GPU (Santos-Martins et al., 2021) as the oracle function. For this task, we use the protein structure from PDB entry 7GAW, where an inhibitor discovered through the COVID Moonshot project (Boby et al., 2023) is co-crystallized with Mpro2. This inhibitor serves as the baseline molecule for evaluation. We set the initialization query to Lipinski's Rule of 5 to generate drug-like molecules as starting candidates and perform genetic algorithm over the query space containing the ECFP4 fingerprint as the optimizable term. 2,000 oracle calls are budgeted. As shown in Figure 4, our method can generate molecules from scratch that achieve improved docking scores compared to the baseline inhibitor. Visualizations of the docking poses further reveal that the generated molecules share binding modes similar to the baseline inhibitor, including fitting into the highlighted subpocket.

## 4 RELATED WORK

Synthesizable molecular design methods can be roughly divided into two categories. The first directly searches the combinatorial space of synthetic pathways (Vinkers et al., 2003; Hartenfeller et al., 2012; Korovina et al., 2020; Gottipati et al., 2020; Horwood & Noutahi, 2020; Bradshaw et al., 2020; Gao et al., 2021; Swanson et al., 2024; Cretu et al., 2024; Seo et al., 2024; Koziarski et al., 2024). However, as discussed in Section 1, the action space is intractably large and valid pathways are sparse, leading to poor sampling efficiency. Common workarounds, such as prefiltering or ranking building blocks, often result in suboptimal solutions. The second category trains models to construct synthetic pathways from input molecular graphs (Luo et al., 2024; Gao et al., 2024; Sun et al., 2024; 2025; Lee et al., 2025). These models, however, cannot generate molecules conditioned on property specifications and remain limited in both chemical space coverage and efficiency. Beyond these two categories, other methods focus on specific problem classes, such as structure-based drug design (Jocys et al., 2024; Rekesh et al., 2025), or formulate the task differently, for instance, treating synthesizability as an optimization objective (Guo & Schwaller, 2024; 2025).

Another line of research relevant to this work is compositional generative models (Du et al., 2020; Du & Kaelbling, 2024), which generate images or texts by combining distributions over different concepts. This approach has also been applied to natural language generation (Liu et al., 2021).

## 5 CONCLUSION

We present PrexSyn, a generative framework for synthesizable molecular design that enables programmable generation of molecules satisfying complex property queries. Combined with the query space–based optimization algorithm, it efficiently samples molecules with respect to black-box oracle functions. Extensive experiments demonstrate PrexSyn's high coverage of the synthesizable chemical space, its effectiveness in complex query-based design, and its strong sampling efficiency, highlighting its potential to further the practical impact of generative AI in molecular design.

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

## A  ADDITIONAL RESULTS

### A.1  COMPLEX PROPERTY QUERYING

Table A1: Examples of generated synthesizable molecules in the complex property query setting.

| Task | Oracle | Pathway and Product |
|------|--------|---------------------|
| Cobimetinib | 0.9398 | |
| Cobimetinib | 0.9321 | |
| Cobimetinib | 0.9278 | |
| Osimertinib | 0.9562 | |
| Osimertinib | 0.9358 | |
| Osimertinib | 0.9314 | |

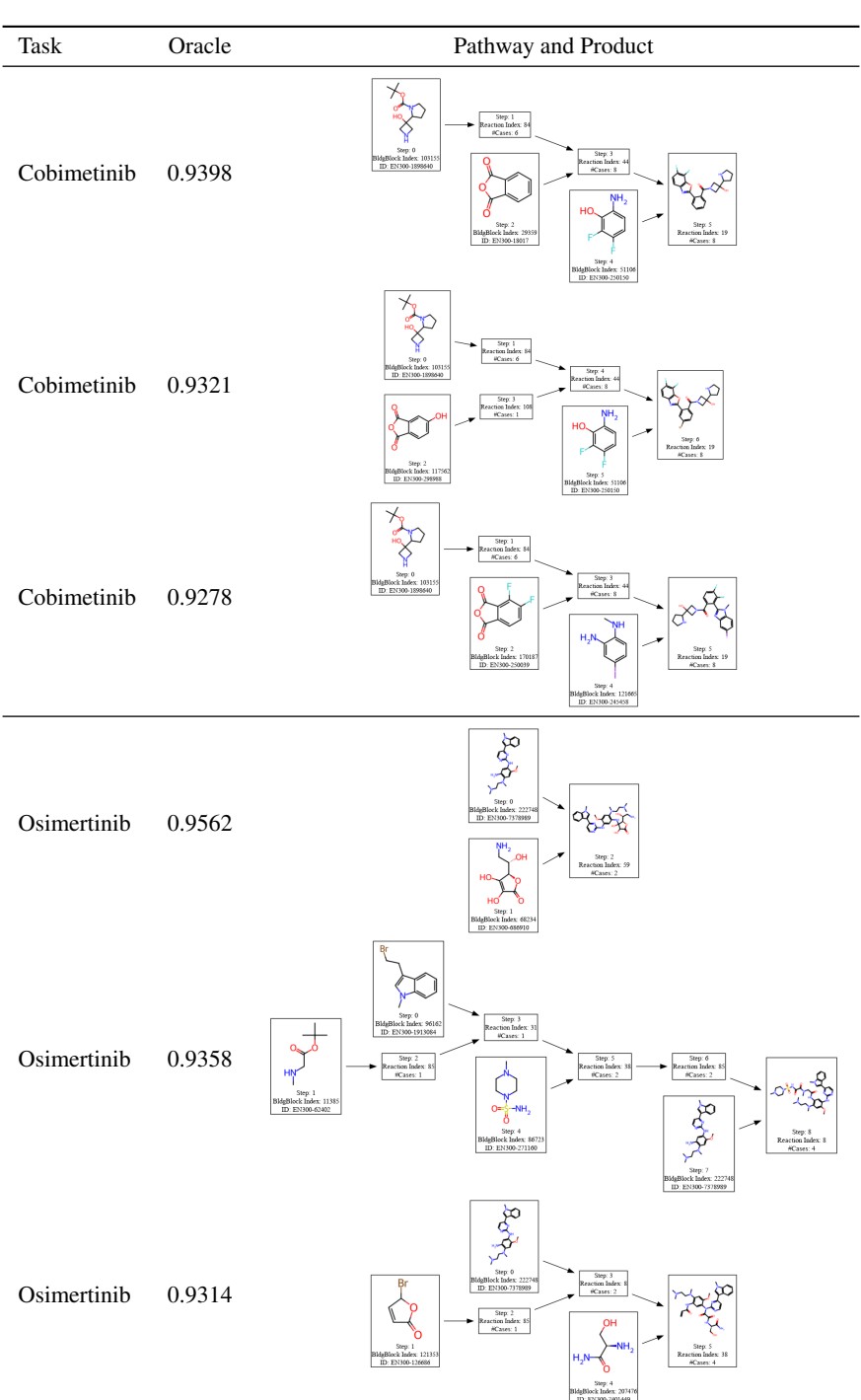

## A.2 Molecular Optimization

We present below visualizations of the optimization landscapes for the scaffold hopping task with and without composite query conditioning (Figure A1), as well as for the Celecoxib Rediscovery and Amlodipine MPO tasks (Figure A2). The 2D projections are generated using the tSNE algorithm (Maaten & Hinton, 2008) applied to the ECFP4 fingerprints of the sampled molecules.

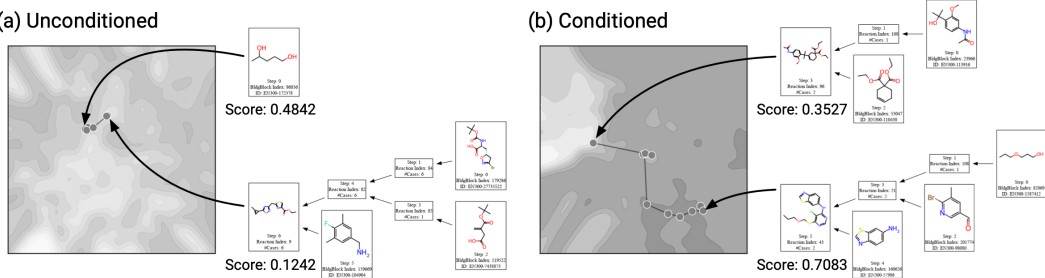

Figure A1: Scaffold hopping optimization landscapes with and without composite query conditioning. **(a)** The unconditioned optimization explores a rugged landscape, where high-scoring molecules are sparse and difficult to locate. The resulting candidates often fail to satisfy the desired decoration constraints. **(b)** The conditioned optimization incorporates a composite query decoration substructures as constraints, leading to a smoother optimization landscape and thus higher-scoring molecules.

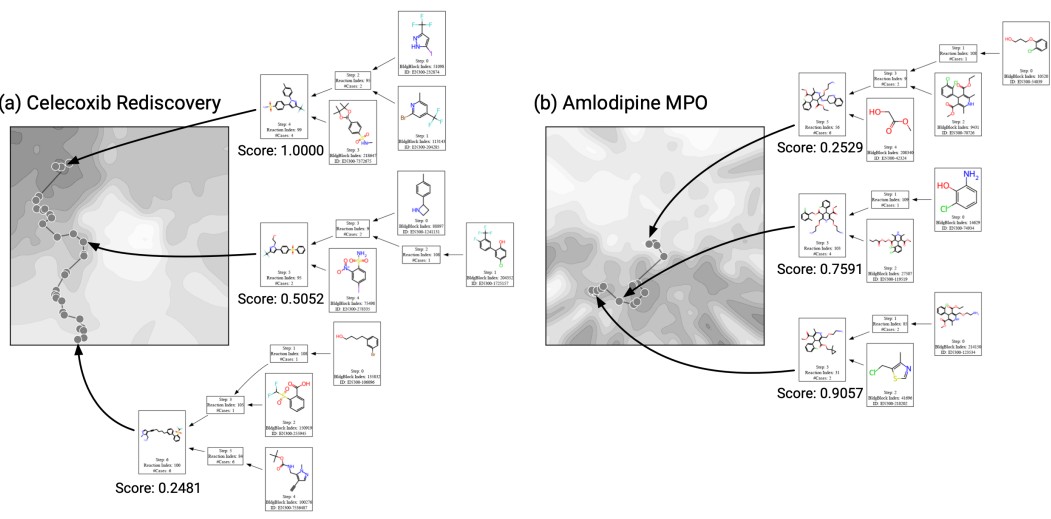

Figure A2: Optimization landscapes of the Celecoxib Residcovery and Amlodipine MPO objectives from the GuacaMol benchmark suite (Brown et al., 2019).

## B Implementation Details

### B.1 Parameters

The transformer architecture used in our experiments consists of 12 layers, with a model dimension of 1024, a feedforward dimension of 2048, and 8 attention heads. We employ the Adam optimizer with a learning rate of $3 \times 10^{-4}$, a cosine annealing learning rate scheduler, and a batch size of 2048. The model is trained for 48 hours on two NVIDIA H100 GPUs, which corresponds to approximately 640,000 iterations.

### B.2 DATA PIPELINE

We randomly construct synthetic pathways on the fly during training, following the approach of ChemProjector (Luo et al., 2024). For each product molecule, property values are computed. In previous works, the data pipeline generated only linear pathways. To increase both the training data diversity and the model capacity, we extend the pathway generation algorithm to produce branched pathways by pre-applying unimolecular reactions to building blocks and storing the intermediates as secondary building blocks, which are then used for pathway generation during training.

### B.3 PRETRAINED PROPERTIES

Table A3: Pretrained molecular properties used in our experiments.

| Name | Description |
| --- | --- |
| `product_ecfp4_fingerprint` | ECFP4 fingerprint of the product molecule |
| `product_fcfp4_fingerprint` | Gobbi pharmacophore-based FCFP4 fingerprint of the product molecule. |
| `product_fragment_fingerprints` | ECFP4 fingerprints of BRICS fragments of the product molecule. |
| `product_scaffold_fingerprint` | ECFP4 fingerprint of the Murcko scaffold of the product molecule. |
| `product_rdkit_properties` | 43 RDKit-calculated molecular descriptors of the product molecule, including average molecular weight (amw), topological polar surface area (TPSA), Crippen logP, number of rotatable bonds, and more. |
| `product_rdkit_property_upper_bounds` | Upper bounds of the 43 RDKit-calculated molecular descriptors. This is used for range queries (*e.g.* `MW<500`). |

### B.4 QUERY COEFFICIENTS

During property composition, the importance of each property can be controlled by tuning its query coefficient (Equations 3 and 5). By default, we set all coefficients to 1. In the Cobimetinib and Osimertinib optimization tasks (Tasks 2 and 3 in Table 2), we instead set the coefficient of the similarity to the reference molecule (`ECFP4(...)`) to 0.75. This encourages the model to explore diverse molecules while still maintaining a certain degree of similarity to the reference. When the coefficient is set to 1, the model tends to repeatedly generate molecules that are highly similar or even identical to the reference molecule.

### B.5 OPTIMIZATION ALGORITHM

Our proposed query space optimization technique is algorithm-agnostic and can be integrated with any optimization method. In our experiments, we use the genetic algorithm which has been widely used in prior works (Jensen, 2019; Tripp & Hernández-Lobato, 2023). Unlike GraphGA that defines evolutionary operations directly on molecular graphs, our method operates only on numerical values, making it more efficient and easier to implement. To cross over two sets of query values, we sample from a two-component isotropic Gaussian mixture model fitted to the parent query vectors. For discrete values (e.g., binary fingerprints), the sampled result is rounded to the nearest integer. For all benchmark tasks, we run optimization until the oracle budget is exhausted. The population size is set to 500, and at each iteration, 50 parents are sampled to generate 50 offspring through crossover and mutation, with a mutation rate of 0.1.

### B.6 DOCKING PROTOCOL

We follow the protocol below for molecular docking using AutoDock GPU (Santos-Martins et al., 2021):

- Define the docking grid box centered on the reference ligand, with a box length of 20 Å.
- Prepare the protein structure by removing waters, cofactors, and ligands, then add hydrogens using REDUCE (Word et al., 1999).
- Use `mk_prepare_receptor.py` (Martins et al., 2025) to convert the prepared protein structure to an AutoGrid file.
- Protonate and generate 3D conformers of the ligand using Molscrub (Diogo et al., 2025).
- Convert the ligand structure to PDBQT format with `mk_prepare_ligand.py` (Martins et al., 2025).
- Perform docking with AutoDock GPU.
- Analyze docking results and select the cluster with the lowest mean binding energy as the final result, using the cluster centroid as the representative binding pose.

