# OpenReview forum: "Programmable Exploration of Synthesizable Chemical Space"
_ICLR.cc/2026/Conference — Submitted to ICLR 2026_

### Official Review · Reviewer_ZmiN · 2025-10-28

**Soundness:** 2
**Presentation:** 2
**Contribution:** 2
**Rating:** 2
**Confidence:** 3

**Summary:**

This paper proposes PrexSyn, a programmable generative model designed to directly explore the synthesizable chemical space.
As input features, it uses descriptors such as ECFP4 fingerprints, molecular weight, ClogP, and TPSA.
In addition, the authors applied a method inspired by logic gates, adopting a product-of-experts formulation.
The model was tested on various benchmarks, including Enamine REAL, ChEMBL, Guacamol, and sEH binding (docking).
They employed an iterative black-box oracle-guided refinement process.

**Strengths:**

1. They tested the model on a wide range of benchmarks.

2. They also proposed a novel logic gate–inspired approach.

**Weaknesses:**

1. PrexSyn uses structural descriptors such as ClogP, TPSA, and molecular weight, which are also used as objectives in benchmarks like GuacaMol.
Since these descriptors are explicitly included during model training, there is a risk of label leakage, where the model indirectly learns test-time objectives.
Consequently, the reported oracle efficiency may not accurately reflect generalization to unseen or functional properties beyond the training distribution.

2. The proposed method relies heavily on black-box oracle feedback for query refinement.
This raises questions about the fairness of comparison, as oracle calls during query perturbations may effectively exceed those in baseline methods.
Using oracle functions also consumes significant computational resources, so it seems reasonable to assume that they would incur a higher computational cost compared to methods like Synformer or ReaSyn.

3. The main table includes ReaSyn, but after checking the ReaSyn paper directly, I found that the experiments related to the sEH binding in Table 4 are also included in the Reasyn paper. This is quite puzzling.

4. The paper claims Logic in its method, but in the end, it’s just a product of probabilities. I don’t think the term “logic” is appropriate in this context.

**Questions:**

See the Weakness.

---

> ### Author Response · Authors · 2025-11-21
>
> We appreciate the reviewer's feedback. Below are our responses.
>
> ---
>
> **[Q1]** Pretraining structural descriptors might cause label leakage on GuacaMol?
>
> - There is **no label leakage**. Our training molecules are generated nearly uniformly and are not biased toward any benchmark objectives; the training data therefore contains no information about the GuacaMol tasks. During optimization we also do not construct queries involving descriptors that appear in the GuacaMol scoring functions.
>
> - To further verify that there is no leakage, we trained a second model without structural descriptors (fingerprints only). As shown below, the performance is nearly identical, confirming that training on these descriptors do not benefit the benchmark tasks.
>
> |                               | Amlo. | Fexo. | Osim. | Peri. | Rano. | Sita. | Zale. | Cele. |
> | ----------------------------- | ----- | ----- | ----- | ----- | ----- | ----- | ----- | ----- |
> | with descriptors              | 0.781 | 0.837 | 0.855 | 0.714 | 0.807 | 0.471 | 0.504 | 0.801 |
> | without descriptors (fp only) | 0.788 | 0.826 | 0.863 | 0.719 | 0.810 | 0.476 | 0.503 | 0.804 |
>
> ---
>
> **[Q2]** Our method relies on black-box oracle feedbacks so it's inefficient and unfair?
>
> - **All the methods (including SynFormer and ReaSyn) rely on black-box oracle evaluations** to guide graph edits during search steps. This is inherent to the task definition.
> - The comparison is strictly fair: as stated in L404, "*all methods are allowed a budget of 10,000 oracle calls*". Within this budget, our approach achieves higher AUC-Top10 (Table 3), showing that it finds high-scoring molecules more efficiently than the baselines.
>
> ---
>
> **[Q3]** Missing comparison to ReaSyn on the sEH benchmark?
>
> - The ReaSyn paper does not provide details about the sEH experiment (e.g., oracle budget, protocol), making direct comparison ambiguous. Additionally, at the time the preprint and initial code release became available (Sep 19, five days before the conference deadline), the repository did not include code to reproduce their reported sEH results.
> - For a fair evaluation, we modified the `optimize_tdc.py` script in the ReaSyn code repo for the sEH oracle and ran ReaSyn under the same 10,000-call oracle budget. Under identical conditions, ReaSyn achieves an average best-1000 sEH score of 0.93, while our method reaches 1.01.
>
> ---
>
> **[Q4]** "Logic" is inappropriate because "it's just a product of probabilities"?
>
> - Our use of "logic" refers to the logical operators (AND, OR, NOT) that define how multiple property constraints are combined. The product-of-probabilities formulation **is the probabilistic implementation** of these logical compositions. In other words, the method performs logical combination at the property level and uses probability operations to realize these logical relations during generation.

---

### Official Review · Reviewer_Ws5Y · 2025-10-28

**Soundness:** 2
**Presentation:** 2
**Contribution:** 2
**Rating:** 2
**Confidence:** 3

**Summary:**

This paper presents PrexSyn, which is a generative framework for designing synthesizable molecules by directly generating synthetic pathways in the form of postfix notation rather than unconstrained molecular graphs. The model uses a decoder-only transformer that takes molecular property prompts as input and autoregressively generates sequences of building blocks and reaction templates from a library of 223,244 purchasable building blocks and 115 reaction templates. To enable complex multi-property molecular design, the authors develop a sampling algorithm that compiles logical queries (using AND, NOT, OR operators) into arithmetic combinations of probability distributions conditioned on individual properties, allowing users to "program" generation objectives. For molecular optimization with respect to black-box oracle functions, the authors introduce a query space optimization approach that iteratively refines property queries based on oracle feedback using genetic algorithms, avoiding direct selection from the vast building block action space.

**Strengths:**

1. The authors employ postfix notations of synthesis using purchasable building blocks and validated reaction templates, which enhances the practical synthesizability of the generated molecules and provides explicit synthetic pathways rather than relying solely on heuristic synthesizability scores.

2. The authors address multi-objective molecular discovery tasks and incorporate multiple property constraints, which are crucial considerations in practical drug discovery scenarios.

3. The use of logical operators (AND, OR, NOT) is an interesting concept

**Weaknesses:**

**1. Limited novelty**
- Postfix notation of synthesis representation is directly taken from ChemProjector (Luo et al., 2024)

- Also, the concept of the query-based molecular optimization technique is not novel, as it appears to be similar to and has already been introduced in the existing work by Hoffman et al. (2021), “Optimizing Molecules Using Efficient Queries from Property Evaluations.”

**2. Missing critical baselines and clarifications**
- There is no comparison with recent strong baselines such as ReaSyn (2025) on the sEH binding task (Table 4). While the authors include ReaSyn in the main results table, it is unclear whether its exclusion from Table 4 was intentional, given that the proposed framework appears to underperform against ReaSyn on this specific task.

- Moreover, the detailed explanation of the differences between PrexSyn (Last) and PrexSyn (Top) remains unclear and somewhat confusing.

- No comparison with graph-based multi-objective optimization methods that could provide synthesizability post-hoc

**3. Insufficient ablation studies**
- No ablation on the choice of structural properties used during training (why these specific properties?)

- No detailed analysis on the impact of different query coefficients (beyond a brief mention in Appendix B.4)

- The choice of the genetic algorithm for query space optimization is not well justified, and no comparison with other optimization strategies is provided.

**4. Scalability concerns**
- The vocabulary size of over 223K tokens can be quite large for a molecular discovery framework; however, no analysis of the associated memory or computational requirements is provided.

- There is no discussion of how the proposed method would scale to substantially larger building block libraries, which is important given that real chemical spaces may contain billions of possible molecules.

**5. Insufficient discussion**
- There is no analysis of which types of property combinations are difficult for the model due to potential property conflicts, nor an explanation of why these difficulties arise.

- There is no discussion of when the query-based generation fails or produces low-quality molecules

**Questions:**

1. The paper’s core component relies on the structure–function principle, and the authors train the model solely on structural properties (e.g., ECFP4, scaffolds, descriptors) but expect it to generalize to functional properties at inference time without providing validation of this capability. However, many crucial drug properties exhibit complex structure–function relationships that may not be well captured by simple fingerprints, raising concerns about how such generalization can be achieved.

2. I also wonder whether the model suffers from mode collapse when optimizing for specific properties

---

> ### Author Response · Authors · 2025-11-21
> **Response (1/2)**
>
> We appreciate the reviewer's feedback. Below are our responses.
>
> ---
>
> **[Q1]** Limited novelty?
>
> - The "query-based optimization" in Hoffman et al. (2021) is **fundamentally different** from our query-space optimization, **despite the superficial similarity in naming**. In Hoffman et al., a *query* denotes an oracle call used to update the latent embedding of a molecular autoencoder. In contrast, in our work a *query* is a structured specification of molecular **properties** (e.g., fingerprints, scaffolds, descriptors) used to guide generation. Our contribution lies in optimizing over an explicit, interpretable **property-query space**, which is irrelevant to Hoffman et al.'s latent-space refinement.
> - We have sufficiently acknowledged the contribution of ChemProjector. Building on their postfix notation representaion, our main contribution is advancing from graph-conditioned projection to complex property-conditioned generation which enables efficient molecular sampling, thereby bridging the gap between molecular properties and synthesizability.
>
> ---
>
> **[Q2.1]** Missing baselines?
>
> - The ReaSyn paper does not provide details about the sEH experiment (e.g., oracle budget, protocol), making direct comparison ambiguous. Additionally, at the time the preprint and initial code release became available (Sep 19, five days before the conference deadline), the repository did not include code to reproduce their reported sEH results.
> - For a fair evaluation, we modified the `optimize_tdc.py` script in the ReaSyn code repo for the sEH oracle and ran ReaSyn under the same 10,000-call oracle budget. Under identical conditions, ReaSyn achieves an average best-1000 sEH score of 0.93, while our method reaches 1.01.
>
> ---
>
> **[Q2.2]** Comparison to methods that provide synthesizability post-hoc?
>
> - Methods that generate molecules first and apply synthesizability filters **post-hoc** suffer from **reduced sample efficiency**: molecules rejected by post-hoc filters still consume oracle calls. By contrast, our approach generates molecules within the synthesizable space by construction, avoiding wasted evaluations.
> - We also emphasize that heuristic measures such as the SA score provide **far weaker** evidence of synthesizability than presenting a full synthetic pathway composed of purchasable building blocks and validated reaction templates, which both our model and other synthesis-aware baselines provide.
>
> ---
>
> **[Q3]** Ablation studies?
>
> - We adopted genetic algorithms (GAs) because they are the **standard optimization strategy used by the baselines** (GraphGA, SynFormer, ReaSyn, ...), enabling fair, controlled comparison. Our framework is fully compatible with more sophisticated algorithms (e.g., GPBO, Genetic GFN), but using them would give our method an unfair advantage and confound whether performance gains arise from model quality or sampling strategy. Since our contribution focuses on improving the **model**, we intentionally kept the algorithm as consistent as possible with previous methods.
>
> ---
>
> **[Q4]** Scalability issues?
>
> - Commercial **building block libraries** contain on the order of **10k–500k** entries (e.g., Enamine 223k, WuXi 90k).
> - **Libraries with billions of molecules are virtual spaces constructed from these building blocks**; they are **not building-block libraries themselves**. Our model is explicitly designed to generate over these combinatorial spaces, and our high reconstruction rate on the Enamine REAL (billion-scale) library demonstrates that our representation and vocabulary scale effectively to such large chemical spaces.
> - Furthermore, our model uses less memory than prior fingerprint-based building-block retrieval methods such as ChemProjector and SynFormer, which store all building-block fingerprints in GPU memory for nearest-neighbor lookup. In fact, SynFormer’s building-block fingerprints alone occupy 1.8 GB, whereas the building-block classifier parameters in our model require only 914 MB. Moreover, our transformer has 131 M parameters (512 MB), compared to SynFormer’s 230 M parameters (899 MB). Overall, our model **uses less memory than previous state-of-the-art approaches**.
>
> ---
>
> **[Q5]** Property conflict issues?
>
> - When properties are contradictory (e.g., requiring molecular weight <100 while having >50 heavy atoms), no model can satisfy all constraints simultaneously.
>
> ---
>
> **[Q6]** Fingerprint is limited?
>
> - To model molecules in computer, one must choose a structural representation for molecules. The options can be fingerprints, strings (e.g. SMILES), or graphs, when modeling without 3D conformations. None of them can perfectly capture functional properties, but we have already shown our fingerprint-based method is significantly better than previous models that are based on SMILES (e.g. SynFormer) or graphs (e.g. ReaSyn), especially in property-guided molecular optimization benchmarks.

---

> > ### Author Response · Authors · 2025-11-21
> > **Response (2/2)**
> >
> > **[Q7]** Mode collapse?
> >
> > - Most benchmark optimization tasks (e.g., Celecoxib rediscovery) have only one uni-modal optima by definition.
> > - For the more open-ended sEH binding task, we observe diversity: the average pairwise diversity (1-Tanimoto similarity over Morgan fingerprint) among the selected molecules is 0.6843. For reference, the diversity reported by a popular structure-based drug generative model Pocket2Mol is 0.688. This indicates that the model does not suffer from mode collapse.

---

### Official Review · Reviewer_MAr7 · 2025-10-31

**Soundness:** 3
**Presentation:** 3
**Contribution:** 3
**Rating:** 6
**Confidence:** 4

**Summary:**

This work proposes PrexSyn: a transformer-based model for property-conditioned synthesizable molecule generation. It works by decoding a synthesis tree description based on a property query prompt. The authors then evaluate this on several benchmarks and show promising results.

**Strengths:**

**(S1)**: Synthesizable molecule optimization is a practical and important topic in ML-assisted drug discovery. Experiments are relatively broad and show generally promising trends.

**(S2)**: The handling of AND property constraints by combining next-token probability distributions stemming from separate decoding rollouts for each constituent property is an interesting setup, which nicely sidesteps the need to train under varying numbers of AND-connected properties. It looks like a neat and novel trick to me, although it is possible that it has already appeared in prior work that I am not aware of.

**(S3)**: The paper is well-written and mostly clear (apart from things mentioned in **(W1)** below).

**Weaknesses:**

**(W1)**: A few aspects of the work are so far not fully clear to me:

- **(W1a)**: The authors mention the ECFP4 fingerprint is a supported molecular property, which would make it a vector-valued property (with the exact length depending on how it's folded). The example in Figure 1c shows a function `ECFP` with a binary string argument, while examples in Table 2 a function `ECFP4` with a SMILES as argument (both suggesting a scalar-valued property of fingerprint _similarity_ rather than fingerprint itself). Am I correct to assume the fingerprint is always a scalar-valued property of fingerprint similarity to some vector, and that vector can be derived as a fingerprint of some given seed molecule, although that can also change later during mutation?

- **(W1b)**: When generating the postfix notation of synthesis, can the model pick any forward reaction template, as long as it matches the reactants (i.e. masking out inapplicable reactions), or is there any validation of applicability, e.g. through a forward reaction model? Template match alone doesn't guarantee the reaction would actually make sense (and in fact it's a somewhat weak signal, because in the forward direction, given reactants, one could say usually a single reaction can only happen, so if two templates match, one of them likely does not occur in real life). Also, what happens if template application produces more than one result (stemming from the template matching in multiple places due to symmetries, e.g. see Figure 2 in [1])? These questions would largely also apply to SynFlowNet, but I'm equally unsure if they are addressed there (please correct me if they are).

- **(W1c)**: Is there any explanation behind the optimization results in Figure 3 being the same up to ~1k steps, and then the conditioned approach rapidly outperforming? It's a minor point, but some intuition behind this would be great.

**(W2)**: The scaffold-based tasks use soft scaffold presence scoring, which is a practical approach to make the optimization tractable, but in real-world drug discovery one would often prefer an exact match of the scaffold (and optimize properties under that constraint). It's fine to optimize the soft constraint, but it would be great to see the results then evaluated under the strict match requirement; I wonder if PrexSyn's optimization is flexible enough to find solutions which match the scaffold exactly. One point of comparison would be the scaffold-based Guacamol-style tasks from the MoLeR paper [2].

---

**Nitpicks**

- "oracle function is defined as the negative docking score predicted by a proxy models" - shouldn't this be "proxy model" (singular)?

**References**

[1] "Chemist-aligned retrosynthesis by ensembling diverse inductive bias models"

[2] "Learning to Extend Molecular Scaffolds with Structural Motifs"

**Questions:**

See **(W1)** above for questions.

---

> ### Author Response · Authors · 2025-11-21
>
> We thank the reviewer for the thoughtful and constructive feedback! Please find below our responses to the reviewer's comments.
>
> ---
>
> **[W1a]** Clarification on ECFP4 conditions
>
> - Our model treats ECFP4 as a vector-valued binary fingerprint. This fingerprint is embedded through a linear layer and used as part of the property prompt.
> - In Table 2, ECFP4("smiles string") denotes the full ECFP4 fingerprint vector computed for the molecule represented by the given SMILES string.
> - The fingerprint property is always the fingerprint vector itself. We don't input a target fingerprint similarity score to the model.
>
> ---
>
> **[W1b-1]** Validity of reaction template matching?
>
> - While template matching does not guarantee full chemical validity in real world, we restrict our templates and building blocks to those provided and validated by chemical vendors (e.g. Enamine, WuXi).
> - Enamine estimates an average experimental success rate of ~85% for synthesizing molecules in REAL space [1]. Similarly, WuXi reports an average success rate of ~75% when using their building blocks and reaction protocols [2].
> - Thus, while perfect synthesizability prediction is unrealistic for any computational method, constraining generation to chemical vendor-validated building blocks and reactions establishes a **reliable lower bound**. This is also more trustworthy than pathways proposed by automated retrosynthesis tools which rely on a broader set of reactions which are not necessarily well-validated.
>
> [1] https://enamine.net/compound-collections/real-compounds
>
> [2] https://www.biosolveit.de/wp-content/uploads/2021/08/Xu.pdf (Page 4)
>
> ---
>
> **[W1b-2]** Dealing with multiple products
>
> - When a template application produces multiple products, all products are carried to the subsequent reaction step. For example, if step one produces A1, A2, A3, and A4, and step 2 (+ building block B) yields 2 product for each intermediate, we will get 8 products: (A1B)1, (A1B)2, ..., (A4B)2.
>
> ---
>
> **[W1c]** Interpretation of the optimization curve at ~1k steps
>
> - In the early stage, both conditioned and unconditioned optimization perform a largely random walk and therefore progress similarly slowly. After ~1k steps, the query-conditioned model discovers a direction in chemical space guided by the substructure condition, allowing it to climb the scoring landscape more efficiently. In contrast, the unconditioned model does not have structural guidance and continues struggling in the uninformative landscape.
> - Figure A1 illustrates this difference: the conditioned landscape is substantially smoother, with high-score regions occupying a larger volume.
>
> ---
>
> **[W2]** Hard structure matching evaluation
>
> - 91.2% of all the sampled molecules contain the two exact substructures highlighted in gray in Figure 3.
> - This is one of the benefits of query conditioning, because the substructure is explicitly encoded in the prompt, the model is biased to generating molecules that preserve the required substructure throughout optimization, aoivding accidental removal or modification of these key structural factors.
>
> ---
>
> Please feel free to let us know if the reviewer has further questions or concerns.

---

### Official Review · Reviewer_aCNv · 2025-11-01

**Soundness:** 2
**Presentation:** 2
**Contribution:** 2
**Rating:** 4
**Confidence:** 4

**Summary:**

This paper presents a decoder only transformer model (PrexSyn) for generating products from a combinatorial search space according to a postfix notation of synthetic pathways based on user specified molecular property queries. Through experiments, the authors demonstrate PrexSyn's ability to perform tasks such as retrieving analogues from the Enamine REAL library, sample compounds subject to logical conditions on property values, and optimize black box property objectives by iteratively refining query embeddings.

**Strengths:**

This paper is well motivated by a need for better computational tools to generate compounds out of structured, synthesizeable chemical spaces that satisfy a set of user-specified requirements, such as drug-likeness. Many of the existing generative models or RL approaches that have been proposed for this problem do not handle the programmable aspect that this paper has focused on, and therefore would require separate models for different constraints. So the ability to handle such user specified requirements without retraining is certainly a strength of this paper. While there exists recent prior work on utilizing decoder only transformer models with molecular properties as prompts towards building programmable generative models, the authors focus on using a postfix notation for constructing products out of combinatorial chemical spaces is original and has practical utility. The approach to combining the predictions compositionally for a set of logical conditions appears unique and interesting.

**Weaknesses:**

I feel that the paper’s main weaknesses lie in its lack of novelty (the model mainly recombines established ideas like product-of-experts conditioning, postfix synthesis representations, and transformer-based conditional generation) and limited empirical validation of the idea's merit. Although I found the exposition interesting, the proposed logical query composition relies on strong (and not realistic) independence assumptions. The synthesizability guarantees are also limited to syntactic feasibility with respect to the postfix notation (which induces a valid SMILES), not necessarily practical synthesis, so the claims around synthesizability are perhaps a bit overzealous. I also found the claims relating to functional generalization to not be convincingly demonstrated, since all the downstream tasks in question are related to docking (i.e., structural, not functional). The experimental evaluations invite questions around the design and key experiments such as those summarized in Table 2 are lacking a baseline for comparison, so the quality of the reported performance is unclear.

**Questions:**

Can you expand on the training setup? How are prompts sampled / constructed during training? It isn't clear to me whether all of the properties are being passed as input to the prompt or a subset of them, how the threshold values are chosen (since prompts like "MOL_WT < 500" can be supplied, that suggests that inequalities were used in the prompts during training), etc. How are the more complex prompts like ECFP, FeatureSim, and ScaffoldSim utilized in training vs. inference? I.e., can a user specify a prompt on ECFP based similarity to a query SMILES that was not utilized during training (or to a scaffold), or do these need to be pre-selected at training time?

Table 1 reports results of different recent algorithms on the chemical space projection task, along with the proposed PrexSyn. However it would be helpful to compare against traditional ECFP-based analog retrieval, which are established in the industry for identifying structurally similar compounds to a query molecule out of a large library of compounds. It would be instructive to know, as a baseline, how would traditional analog retrieval perform on these reconstruction tasks relative to PrexSyn? I ask because the reconstruction rates for the baselines considered in the paper are very low on Enamine REAL in contrast to what would be expected from an analog retrieval.

In Table 2, the results are reported in terms of the average of the percentage of constraints satisfied correctly. In many contexts, violation of even a single constraint can be unsatisfactory. It would be helpful to also report on the percent of compounds that satisfy all constraints (i.e., where even a single constraint violation deems a compound as negative).

---

> ### Author Response · Authors · 2025-11-21
> **Response (1/2)**
>
> We thank the reviewer for the thoughtful and constructive feedback! Please find below our responses to the reviewer's comments.
>
> ---
>
> **[W1]** Novelty?
>
> - Our core contribution is an advance in practical capability: we show that the integration of the components enables a near-perfect reconstruction of synthesizable chemical space and substantially higher molecular sampling efficiency .
>
> ---
>
> **[W2]** Claim of synthesizability?
>
> - Our synthesizability claim is grounded in chemical vendor-validated chemistry: we use only purchasable building blocks and experimentally validated reaction templates from a specific chemical vendor (Enamine).
> - Enamine estimates an average experimental success rate of ~85% for synthesizing molecules in the REAL space[1]. Similarly, WuXi reports an average success rate of ~75% when using their building blocks and reaction protocols[2].
> - Thus, while perfect synthesizability is unrealistic for any computational method, constraining generation to chemical vendor-validated building blocks and reactions establishes a **reliable lower bound**. This is also more trustworthy than pathways proposed by automated retrosynthesis tools which rely on a broader set of reactions which are not necessarily well-validated.
>
> [1] https://enamine.net/compound-collections/real-compounds
>
> [2] https://www.biosolveit.de/wp-content/uploads/2021/08/Xu.pdf (Page 4)
>
> ---
>
> **[W3]** Generalization to functional properties?
>
> - In fact, all computational oracles ultimately consume molecular structures. Even direct functional property predictors (e.g. antibiotic activity [3]) require structural representations as input. Thus, the boundary between "structural" and "functional" properties in computational settings is inherently blurred.
> - We find docking among the closest available approximations to functional activity. We also evaluated our model with a bioactivity predictor (DRD2 from TDC benchmark), where our model achieved **0.96 out of 1.00 AUC-Top10** under 10,000 oracle calls budget. However, these predictors are regressors on 2D fingerprints and are generally easy to optimize. In contrast, docking provides a harder challenge that better reflects the intended notion of functional generalization.
>
> [3] A deep learning approach to antibiotic discovery. Cell 2020.
>
> ---
>
> **[W4]** Baselines for Table 2 (composite property queries)?
>
> - For Tasks 2 and 3 we included baselines in the main text (Lines 350–357). Although the settings of the baseline is different, they demonstrate that our query-based approach performs competitively and does not underperform relative to either white-box or black-box optimization methods.
>   - *"For comparison, a graph-based approach that explicitly optimizes the four conditions, rather than treating the scoring function as a black box, reported a top score of 0.93 (Verhellen, 2022)"*
>   - *"While the strongest baseline reported by Brown et al. (2019) achieved a slightly higher top score of 0.95, their method directly modifies molecular graphs without guaranteeing synthesizability."*
> - Tasks 4 and 5 are primarily demonstrations of capability. Their purpose is to illustrate how the querying capability can be applied to docking.
>
> ---
>
> **[Q1]** Property-conditioning details
>
> - **Single-property training:** Each training sample contains *one* property type paired with a synthetic pathway. This avoids exponential training data requirements that arise from enumerating all property combinations[4], and motivates the need for logical composition during inference.
> - **Upper-bound properties (e.g., MW < 500):** We train a separate embedding layer for upper-bound conditions by sampling random threshold values above the molecule’s true property value.
> - **Representations of ECFP, FeatureSim, and ScaffoldSim:** These are represented using (i) ECFP4 fingerprints, (ii) Gobbi pharmacophore fingerprint, and (iii) ECFP4 of the scaffold, respectively. Each fingerprint is embedded via a linear layer into the prompt.
> - **Generalization: **Users may provide fingerprints of **any** SMILES string / molecular graph, even if unseen during training. Generalization on the ChEMBL projection task supports this: many ChEMBL molecules are not reachable via Enamine building blocks, yet the model can still generate analogs with high similarity.
>
> [4] Compositional generative modeling: A single model is not all you need. ICML 2024.

---

> ### Author Response · Authors · 2025-11-21
> **Response (2/2)**
>
> **[Q2]** Comparison to traditional fingerprint-based retrieval?
>
> - We would like to clarify that our goal is to generate from the exponentially large combinatorial space spanned by building blocks and reactions, not to retrieve from the existing finite library.
> - Thus, reconstruction rate serves as a sanity check, not a performance metric. A model claiming to cover an infinite chemical space should at minimum reconstruct well a known finite subset. The >90% reconstruction rate provides strong evidence that PrexSyn indeed captures the underlying combinatorial space reasonably well, at least much better than all the previous models which show much lower reconstruction.
> - We would expect that fingerprint-based analog retrieval would achieve nearly 100% reconstruction, since it retrieves compounds directly from a finite library.
>
> ---
>
> **[Q3]** Hard-constraint evaluation?
>
> - Task 1: Top-5% and top-10% both achieve 100% strict satisfaction (score = 1.00).
>
> - Task 2: (RotatableBonds = 3, AromaticRings = 3, CLogP < 5.0) Top-5%: 86% satisfy all constraints. Top-10%: 77% satisfy all constraints.
>
> - Task 3: (NOT TPSA < 100 AND CLogP < 1.0) Top-5%: 55% strict satisfaction. Top-10%: 60% strict satisfaction.
>
> - Tasks 4 and 5 do not include hard constraints.
>
> ---
>
> Please feel free to let us know if the reviewer has further questions or concerns.

---

### Meta-Review · Area_Chair_5hGa · 2025-12-15

**Summary:**

This paper proposes Prexsyn for postfix sysnthesis programs conditioned on molecuar property prompts.  It also supports logical queries over properties.

The reviewers generally like the direction, especially when the paper reports a strong project coverage. The decision is mainly driven by the conerns from the reviewers on the limited novelty vs. prior work, uneven baselines and sbaltions. Some reviewers question how strong the “synthesizable” and “programmable logic” claims really are in practice.

I also checked there are no ethic review need.

**Reviewer Concerns:**

Overall feedback was mixed but leaning negative. At least two reviews are firmly negative in the initial round.

Main concerns:

1. Some reviewers feel this is an integration of exisiting ideas (postfix snthesis rep, conditional generation, etc.). In my personal view, it is ok and reasonable to integrate.

2. Baselines + eval gaps: reviewes are not fully convinced on baselines in some tables (esp. constraint/query tasks).
they also requests for stricter metrics (hard satisfaction, not avg. satisfaction), and some confusion around sEH comparisons to ReaSyn.

3. scalabilty is clease something should be reported on the memory/runtime while scaling to the larger space.

**Reviewer Scores:**

Since full discussion did not happen, here is my best guess if reviewers had time to react to the rebuttal:

Reviewer aCNv (score 4): I think this goes up a bit as the authors provide the training details and other strict constrined numbers.

Reviewer MAr7 (score 6): may stay the same as it is already positibe and only ask for some clarification questions

Reviewer Ws5Y (score 2) and Reviewer ZmiN (score 2) may not change too much; or even improve to four.

I expect the discussion to move the paper slightly upward, but it will likely still fall below the bar due to remaining novelty/eval concerns.

---

### Decision · Program_Chairs · 2026-01-26

Reject